# Correlation Analysis between Wheat Flour Solvent Retention Capacity and Gluten Aggregation Characteristics

**DOI:** 10.3390/foods12091879

**Published:** 2023-05-02

**Authors:** Wei Gong, Xiaohua Wang, Fengjiao Wang, Jinshui Wang

**Affiliations:** College of Biological Engineering, Henan University of Technology, Zhengzhou 450001, China; gw1754163531@163.com (W.G.); wang0124xh@163.com (X.W.); w1751323@163.com (F.W.)

**Keywords:** wheat flour, solvent retention capacity, gluten, gluten aggregation characteristic

## Abstract

Solvent retention capacity (SRC) is a test for the solvation of wheat flour. Its functional contribution was predicted according to the swelling behavior of different diagnostic solvents to different polymeric components of wheat. Ten commercial wheat flour varieties were used as raw materials in this study. The flour quality, gluten aggregation and solvent retention capacity, and their correlations were analyzed. The results showed that protein content, wet gluten content, dry gluten content and the swelling index of glutenin were positively correlated with torque maximum (BEM), torque 15 s before maximum torque (AM), torque 15 s after maximum torque (PM) and gluten aggregation energy (AGGEN). Moreover, they were significantly correlated with the solvent retention capacity. BEM, AM, PM and AGGEN were positively correlated with standard solvent water-SRC (WSRC) and lactic acid-SRC (LASRC). For supplemental solvents, ethanol-SRC (EthSRC) was positively correlated with AGGEN. Sodium dodecyl sulphate-SRC (SDSSRC) was highly correlated with peak maximum time (PMT). Metabisulfite-SRC (MBSSRC) and MBS + SDSSRC were also significantly correlated with BEM, AM, PM and AGGEN sodium metabisulfite. There were significant correlations between gluten aggregation characteristic, standard SRC solvent and supplemental solvent. This study provides a theoretical basis for the evaluation of wheat flour quality.

## 1. Introduction

Wheat flour plays an important role in the daily human diet, providing essential energy of the body [1]. As the main raw material, it is widely used in various wheat based-flour products [2]. The quality and functional components of flour are the key factors that determine high-quality baked foods [3]. Flour quality is usually assessed using three methods: the composition determination test, rheological test and baking test [4]. The results of composition determination were consistent with the rheological and baking results. The rheological properties of the dough affect end-use characteristics of the flour. Therefore, rheological analysis assesses the performance of the dough. The third stage of the baking function test performs a further evaluation. However, the rheological test results reflect the combined and cumulative contribution of the functional components of the flour rather than assess the impact of individual flour polymers on dough function [5]. Furthermore, Duyvejonck et. al. [6] suggested that different functional compositions of flours could lead to different qualities of the processed final product. This phenomenon indicated that individual functional ingredients were responsible for quality differences.

To some extent, the solvent retention capacity (SRC) (AACC method 56-11.02) had been proposed as a specific diagnostic tool for predicting individual functional components of flour. Briefly, the SRC was a method to assess the swelling behavior of flour polymerization components in the presence of excess diagnostic solvents. The four standard solvents for the SRC method were water, 5% *w*/*w* lactic acid, 5% *w*/*w* sodium carbonate and 50% *w*/*w* sucrose. Of these, water (for all hydrophilic constituents) and 5% *w*/*w* lactic acid solution (for glutenin) were closely related to wheat protein. The four SRC solvents were considered ideal for evaluating the quality of soft wheat flour. They could not be used to evaluate the function of gluten in durum wheat flour because the individual contribution of the gluten subunit was not fully evaluated in a 5% lactic acid solution [2]. In addition to the above four solvents, four new solvents have been proposed, i.e., 0.006%-sodium metabisulfite (for gluten), 0.75%-sodium dodecyl sulphate (for glutenin macropolymer), 0.75%-SDS + 0.006%-MBS (for GMP without S–S network) and 55%-ethanol (for gliadins) [7]. Gliadin is soluble in aqueous-ethanolic solutions and glutenin is soluble in dilute acid solutions. However, the SRC method did not extract solutes dissolved in the supernatant but rather targeted the expansion of a single functional component. This method determined which functional component in the flour affected the swelling behavior of the flour. Notably, MBS (sodium metabisulfite) is a reducing agent used in industry. It was able to reduce and break the gluten disulfide between glutenins and promote free sulfhydryl/disulfide bonds (SH/SS) exchange reactions in strong gluten flours [7]. Therefore, these supplemental solvents were used to study the overall strength properties of glutenin, glutenin polymer (GMP), GMP without S-S network, and wheat gliadin.

To date, a number of studies had used standard solvents to predict the quality of soft and durum wheat flours [8,9]. The relationship between supplemental solvents retention capacity and wheat flour aggregation characteristics has not been reported. In this study, ten kinds of commercial wheat flours were used as raw materials to investigate the wheat flour quality, gluten aggregation characteristics, standard and supplemental solvent retention capacity and their correlations. The purpose of this study was to explore the relationship between the supplemental solvents and the aggregation properties of gluten. It will provide a theoretical basis for the evaluation of wheat flour quality.

## 2. Materials and Methods

### 2.1. Materials

The samples were different gluten wheat flour sold by COFCO brand, Meimei brand, Jinxiang brand, Xiangxue brand and Zhongyu brand. Ten samples were arranged from high to low protein content and stored at 4 °C until use.

### 2.2. Flour Test

Moisture content was determined gravimetrically according to the AACC 44-15.02 method [10]. Wheat flour (2 g) was dried at 130 °C and the moisture content was calculated. The difference of moisture content between the two repeated tests was within 0.2%. The content of wheat flour protein was determined by automatic the Kjeldahl nitrogen determination system. The crude protein content of wheat flour was measured according to the AACC 46-12.01 method [11].Wheat flour (10 g) was added to the automatic gluten wash room. The wet gluten content was obtained after weighing the wet gluten. The wet gluten was centrifuged in the gluten index cassette, and the gluten index was calculated according to the ratio of the gluten to the total weight of the gluten on the sieve. Wet gluten, dry gluten and the gluten index were determined according to the AACC 38-12.02 method [12]. Sedimentation test for flour was performed according to the AACC 56-60.01 method [13]. Flour (3.2 g) was placed in a 100 mL glass-stoppered graduated cylinder and 50 mL of deionized water containing bromophenol blue was added. Flour and water were fully mixed 12 times in 5 s by alternately moving the barrel horizontally and vertically. Flour should be completely swept into suspension. Then 25 mL isopropyl alcohol lactic acid solution was added into glass-stoppered graduated cylinder and mixed for five minutes. Finally, the precipitate value was calculated according to the volume of the precipitate.

### 2.3. Swelling Index of Glutenin Test

The swelling index of glutenin (SIG) was determined by the method of Wang et al. [4]. Lactic acid (85% 10 mL) was diluted to 90 mL and was standing for 24 h to obtain a lactic acid working solution. The 3% SDS-lactic acid solution was obtained by dissolving 30 g of 99% recrystallized SDS in 970 mL of water and then adding 20 mL lactic acid working solution to fix the volume to 1000 mL and kept standing for 24 h. Flour (1.0 g) was weighed into a 50 mL centrifuge tube and 15 mL of water was added. The flour-solvent suspension was vortexed for 5 s until suspended, then kept standing for 10 min at 25 °C and vortexed for 5 s every 5 min. Then, the SDS-lactic acid solution (15 mL) was added into a 50 mL centrifuge tube. The samples were vortexed for 5 s and kept standing for 20 min. Centrifuge tubes were vortexed for 5 s at 5 min intervals during this time. The suspended samples were then centrifuged at 450× *g* for 300 s, then the supernatant was pipetted. Then samples were centrifuged at 450× *g* for another 180 s, then the supernatant was aspirated. The tube weight was determined, and the SIG value was calculated as a percentage of the initial flour weight on a 14% moisture basis.

### 2.4. Disulfide-Sulfhydryl Analysis

Free sulfhydryl (SH) and disulfide bond (SS) were measured according to the method reported by Wang et al. [14]. Flour was added to a 10.0 mL reaction buffer (0.2 M pH 8.0 Tris-Gly with 8 M urea, 3 mM EDTA and 1% SDS). Vortex mixer was used to rotate the suspension at high speed for 1 min. The sample was then vibrated for 60 min at room temperature (250 r/min) and centrifuged for 15 min at 10,000 g. The supernatant was collected for the determination of free sulfhydryl and disulfide bonds. For determination of free SH group content, the supernatant (4 mL) was added with 0.1 mL of 10 mm DTNB, and was oscillated at room temperature (250 r/min) for 3 h. UV photometer was then used to determine the absorption value of the sample at 412 nm. The SH was calculated according to Equation: *C_SH(free)_(μmol/g) = A/εb* [15]. *A* is the absorbance value, *ε* is the extinction coefficient of 13,600, and *b* is the cell path length. For the measurement of total SH contents, the 1 mL supernatant was added with 0.05 mL β-mercaptoethanol and 4 mL of 0.2 M Tris-Gly buffer. Vortex mixer was used to rotate the suspension at high speed for 1 min. The sample was then vibrated for 60 min (250 r/min), adding trichloroacetic acid (13%, *w*/*v*, 10 mL) and then vibrating for 60 min. The precipitates were collected by centrifugation for 15 min at 10,000 g. The pellet was washed twice with 6 mL of 13% trichloroacetic acid (13%), adding 10 mL of 0.2 M Tris-Gly buffer) and 0.1 mL of 10 mM DTNB(5,5′-dithiobis-2-nitrobenzoic acid). The absorbance was determined at 412 nm by UV photometer. The SS group content was calculated according to Equation: $C_{SS}(\mu mol/g)=(C(SH_{total})-C(SH_{free}))/2$.

### 2.5. Gluten Aggregation Testing in GlutoPeak

Gluten aggregation was determined according to the method of Wang et al. [16] with slight modifications. Flour (8.5 g) was dispersed in water (9.5 g), and the combined water and flour was adjusted based on 14% moisture to keep the solid-liquid ratio constant (equal to 1.26). The sample temperature was kept constant at 35 °C, using the jacketed sample cup circulating water, and the paddle speed was set to 2700 r/min and the test time was 240 s. Evaluation of the metrics was performed using the instrument’s software.

Peak maximum time (PMT) corresponds to the time required to reach maximum torque; expressed in seconds. Torque maximum (BEM) corresponds to maximum torque of gluten; expressed in GPU. Torque 15 s before maximum torque (AM) corresponds to the torque value 15 s before maximum torque; expressed in GPU. Torque 15 s after maximum torque (PM) corresponds to the torque value 15 s after maximum torque; expressed in GPU. The GlutoPeak aggregation energy (AGGEN) is automatically calculated from the area of the curve obtained with the device; expressed in cm^2^.

### 2.6. Solvent Retention Capacity

Solvent retention capacity (SRC) of flour samples were determined according to the AACC 56-11.02 [17]. The flour sample (5 g) was dispersed in 25 mL solution (water, lactic acid (5%)). Samples were placed in a thermostatic oscillator (1400 rpm, 5 min, 25 °C) and then centrifuged (4000 g, 2 min). Each precipitation obtained was weighed and the SRC value was calculated. In addition, the supplemental SRC was based on the approach of Kweon et al. [7]. The retention power of supplemental solvents was determined based on the proposed four solvents: 0.006% sodium metabisulfite (MBS), 0.75% sodium dodecyl sulfate (SDS), 0.75% SDS + 0.006% MBS, and 55% ethanol (Eth). The supplemental solvent retention test was also carried out in accordance with the above-mentioned methods.

WSDS: water-SRC, LASRC: lactic acid-SRC, EthSRC: ethanol-SRC, SDSSRC: sodium dodecylsulphate-SRC, MBSSRC: metabisulfite-SRC, SDS + MBSSRC: Sodium dodecylsulphate + metabisulfite-SRC.

### 2.7. Statistical Analysis

Each experiment was repeated more than three times. The data presented are averages of measurements ± standard deviations. The results were compared with the LSD test at *p* ≤ 0.05. The correlations between the qualities of the 10 samples were determined using the Pearson correlation coefficient method (*p* ≤ 0.05 (*) and *p* ≤ 0.01 (**)). Principal component analysis (PCA) was also performed using Origin 18.

## 3. Results

### 3.1. Flour Physical Properties

Table 1 provides the crude protein content of wheat flour, which is concentrated in the range of 5.40~13.08%. The samples were classified according to protein content. Sample 1 was high-gluten wheat flour, samples 9 and 10 were low-gluten wheat flour and samples 2~8 were medium-gluten wheat flour. The gluten network is formed by mixing glutenin and gliadin with water [18]. The quantity and quality of gluten affect the final processing quality of wheat flour. Compared with medium-gluten flour, high-gluten flour had higher wet gluten content, dry gluten content and swelling index of glutenin. The gluten content, dry gluten content and swelling index of glutenin decreased with the decrease in wheat flour protein content. The gluten index reflects the quality of gluten in the dough [19]. The quantity and quality of gluten are of equal importance as two independent factors in the evaluation of wheat flour. The gluten index of wheat flour samples did not decrease with the decrease in crude protein content and in dry and wet gluten content. On the contrary, the lowest crude protein content of sample 8 and sample 9 had higher gluten indexes, indicating that there was no correlation between gluten index and gluten content. The swelling index of glutenin reflects the swelling characteristics and quality of glutenin based on the difference of swelling power of wheat flour in SDS lactic acid solution [20]. The swelling index of glutenin increased with the increase in crude protein content in wheat flour. Wessels et al. [8] also reported a positive correlation between glutenin swelling index and protein content. Sedimentation index ranged from 34.60 mL to 53.05 mL. The sedimentation index is a comprehensive index that reflects the quantity and quality of wheat flour protein. The sedimentation index was positively correlated with gluten strength.

### 3.2. SH and SS Content

Change of free sulfhydryl (SH) and disulfide bonds (SS) content plays a vital role in wheat gluten polymerization [14]. When disulfide bonds in a protein are reduced to free sulfhydryl groups, the protein structure becomes disordered and the structural stability of the protein is destroyed. The content of sulfhydryl disulfide bonds in different wheat flour samples are showed in Table 1. On the whole, the sulfhydryl group content of wheat flour was concentrated in the range of 2.94~4.68 umol/g. The sulfhydryl group content of sample 6 was the highest and sample 3 was the lowest. The higher sulfhydryl content wheat flour had the worse protein network structure. The disulfide bond content of wheat flour was concentrated in the range of 3.58~4.04 umol/g. The disulfide bond content of sample 1 was the lowest and sample 4 was the highest. The higher disulfide bond content reflected that the protein network structure formed by the sample was more stable.

GlutoPeak is a rapid shear-based method that can measure the aggregation behavior of gluten to assess wheat flour quality [21]. In the experiment, the flour and solvent were mixed to form flour paste. Glutenin and gliadin aggregated to form a gluten network. The gluten aggregation curve increased with the formation of the gluten network, and the gluten network was destroyed by further external force after the torque curve reached the maximum value [22]. The peak value of the curve is BEM and the correspond time is PMT [23]. Some earlier studies indicated that PMT was proportional to the formation time of gluten network, and gluten aggregates were formed rapidly in high-gluten wheat flour, showing longer PMT [24]. Lower gluten strength wheat flour had shorter PMT. The higher content of gliadin in low-gluten wheat flour inhibited the development of the gluten network, prolonged the time needed for the completion of the gluten network, and no peaks even occurred [25]. Because samples 9 and 10 were low-gluten wheat flour, no peak value appeared during the test. Karaduman et al. [25] reported that the consistency of low-gluten wheat flour increased rapidly during testing and then decreased rapidly with shorter PMT. On the contrary, higher gluten wheat flour takes longer to peak. As shown in Figure 1, the range of PMT was from 76.25 s to 156 s, and the peak value of gluten was shifted to the left. Sample 2 and sample 8 had longer PMT, indicating higher gluten strength. BEM reflects the force required to destroy the gluten structure, and a higher BEM value indicates the greater force necessary to destroy the gluten [26]. The higher BEM comes from wheat flour with higher gluten strength. BEM was positively correlate with crude protein content except for wheat flour samples without peak value. Kaur Chandi et al. [27] reported that protein content in wheat flour was positively correlated with BEM. It was consistent with the results of this study. The BEM value is also correlated with dry, wet gluten content and the gluten index of wheat flour.

AM is the torque to reach the first 15 s of BEM. It represents the point at which gliadin and wheat gluten start to polymerize, i.e., the initial formation of gluten network. After reaching BEM, the gluten network is broken and the torque value decreases. The value of 15 s after BEM is called PM. AM reflects the gluten strength of flour before the formation of the gluten network, and it is the best index to predict the volume of bread [25]. The AM of wheat flour was 12.25~27.25 GPU. The AM of sample 1 was the highest and sample 7 was the lowest. AM was highly correlated with protein content, dry gluten content, wet gluten content and the glutenin swelling index. PM of wheat flour ranged from 34.00 GPU to 54.25 GPU. Similarly, PM was positively correlated with protein content, dry gluten content, wet gluten content and the glutenin swelling index. AGGEN represents the area from the start of the test to the peak curve. Rakita et al. [28] reported a positive correlation between AGGEN and gluten strength. The low-gluten flour had lower AGGEN value and was suitable for making biscuits. According to Table 2, the AGGEN values of wheat flour samples were 1012.59–1689.25 cm^2^. In addition, AGGEN was positively correlated with protein content, wet, dry gluten content and the swelling index of glutenin.

### 3.3. SRC Assessment

The SRC value can determine the effect of different functional components on flour. Moreover, the SRC value can reflect the quality characteristics of flour. For standard solvents, the gluten properties of wheat flour can be evaluated using the LASRC value. The higher the LASRC value in a certain range, the better the soft gluten quality [29]. Kweon et al. [7] reported that LASRC was a good index for evaluate G’ and G’’ gluten parameters. There was a positive correlation between the SRC value of lactic acid and protein content, which is understandable given the glutenin content in wheat flour [30]. The WSRC reflected the comprehensive properties of wheat flour. Ram et al. [31] reported that the WSRC was positively correlated with wheat flour protein content. Table 3 showed the SRC values of wheat flour. Sample 1 had the largest WSRC and LASRC values. Sample 9 and 10 contained the worst polymer components and showed the lowest WSRC and LASRC values. There was a significant positive correlation between WSRC and protein content. In addition, the obtained WSRC value was always smaller than the LASRC value. Hammed et al. [5] came to the same conclusion on the solvent retention study of wheat flour in different varieties and regions. The LASRC values of ten samples ranged between 80.18% and 139.97%. Moreover, it was positively correlated with wet gluten content, dry gluten content and the glutenin swelling index (Table 4). The positive correlation of LASRC with wet gluten content and glutenin swelling index was also demonstrated in studies by Labuschagne [32] and Wessels [8].

Supplemental solvents had been identified and expanded based on polymer solubility, solvent similarity of gluten, and gluten components. The values of EthSRC, MSSRC, SDSSRC and MBS + SDSSRC were 53.5~73.15%, 52.13~71.14%, 77.01~106.76% and 53.75~74.94%, respectively. In accordance with the results of standard solvents, EthSRC, MSSRC and SDS + MSSRC of sample 1 were the highest among ten wheat flour samples, which were 67.32%, 73.73% and 74.94%, respectively. Samples 3 and 9 had the lowest EthSRC. A total of 55% EthSRC was related to the composition of gliadin [33]. Wheat flour with high EthSRC values may exhibit stronger viscoelasticity [34]. The SDSSRC reaction glutenin macropolymer integrated properties. Sample 8 showed a higher SDSSRC value of 106.76%, indicating the highest glutenin macropolymer strength. The complementary solvent related to the overall strength of gluten was MBSSRC. As a reductant used in industry, MBS can reduce and destroy disulfide bonds between glutenin and promote sulfhydryl/disulfide (SH/SS) exchange reaction in wheat flour [7]. MBSSRC and SDS + MBS SRC values were positively correlated, and the sample with the highest SRC values were consistent. The results indicated that the reduction of the disulfide bond by reducing the agent MBS affected the solvent retention of glutenin macromolecular components.

### 3.4. Relationship between SRC Solvent and Flour Aggregation Properties

WSRC and LASRC were closely related to gluten strength of wheat flour. According to Table 4, the standard solvents WSRC and LASRC were positively correlated with BEM, AM, PM and AGGEN. There was a good correlation between the aggregation characteristics of WSRC and LASRC. The study reported that the extensibility of dough decreased and the gluten strength increased with the increase in the ratio of glutenin to gliadin in wheat flour [35]. HPLC analysis showed that SDS-unextractable glutenin macromer was positively correlated with SDSSRC [36]. In this study, SDSSRC was significantly correlated with PMT, which was consistent with the results reported above. There was a significant positive correlation between AGGEN and EthSRC. The higher the value of EthSRC, the higher the strength of gliadin and the greater the viscoelasticity of dough. Gliadin could delay the peak torque of dough and affect the viscosity of dough during the mixing process. Sladana et al. have reported that wheat flour with higher tenacity has higher AGGEN values in the Glutopeak aggregation test [28]. This is consistent with the results of this study. WSRC, MBSSRC and MBS + SDSSRC were positively correlated with the aggregation characteristics of BEM, AM, PM and AGGEN, with correlation coefficients ranging from 0.725 to 0.919**. These results demonstrated the correlations between the solvent retention capacity and dough aggregation characteristics of wheat flour. It also showed that supplementary solvents could also predict the quality of wheat flour.

### 3.5. PCA

In order to consider all data simultaneously, principal component analysis was performed on the test data of the wheat flour. The variance was 71.8% for PC1 and 13.3% for PC2 (Figure 2). In the PC1 component, protein content, wet gluten content, dry gluten content, swelling index of glutenin, PMT, AM, PM, BEM, AGGEN, WSRC, EthSRC, MBASRC and SDS + MSSRC were closely located in the first and second interval. PC1 component reflected gluten quality, which in turn improved wheat flour quality parameters and gluten aggregation properties. LASRC was in the upper right corner, away from the cluster. Regarding PC2 components, gluten index and SDSSRC were closely located in the third interval. PC2 component reflected gluten strength, SDSSRC and DI in the third interval, showing a positive correlation with gluten strength. The vectors protein content, wet gluten content, dry gluten content, swelling index of glutenin, PMT, AM, PM, BEM, AGGEN, WSRC, EthSRC, MBASRC and SDS + MSSRC form a cluster, showing a positive correlation. SDSSRC and gluten index were located in the upper left of the graph, indicating that they have a positive correlation and behave differently from their vectors, indicating that they were not correlated with the index of the first interval. The results of principal component analysis were consistent with those of correlation analysis.

## 4. Conclusions

In this study, the gluten aggregation characteristics in wheat flour with different gluten strength, the retention capacity of standard and supplemental solvents and their correlations were analyzed. The SRC diagnostic solvent and gluten aggregation characteristics were used to predict the gluten characteristics of wheat flour, and their correlations were analyzed. The results showed that wheat flour with higher protein content, wet gluten content, dry gluten content and swelling index of glutenin had higher BEM, AM, PM and AGGEN values. BEM, AM, PM and AGGEN were correlated with standard solvents WSRC and LASRC. The supplemental solvent EthSRC was associated with AGGEN. SDSSRC was highly correlated with PMT. MBSSRC and MBS + SDSSRC were also significantly associated with BEM, AM, PM and AGGEN. It was concluded that the standard SRC and supplemental solvents were significantly correlated with gluten aggregation characteristics, which provided a theoretical basis for wheat flour quality evaluation.

## Figures and Tables

**Figure 1 foods-12-01879-f001:**
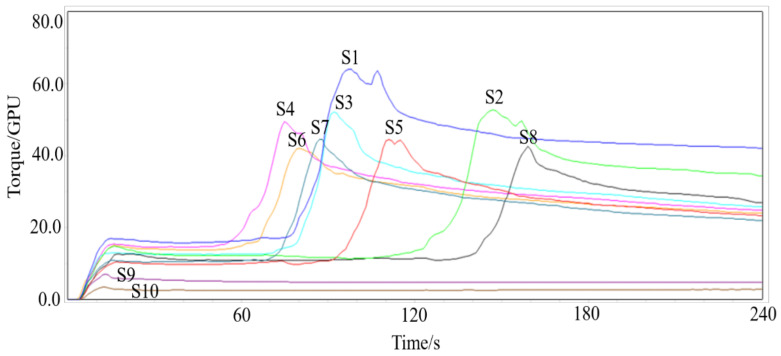
Gluten aggregation characteristic curve of wheat flour. Samples 1–10 are represented in the figure by S1–S10.

**Figure 2 foods-12-01879-f002:**
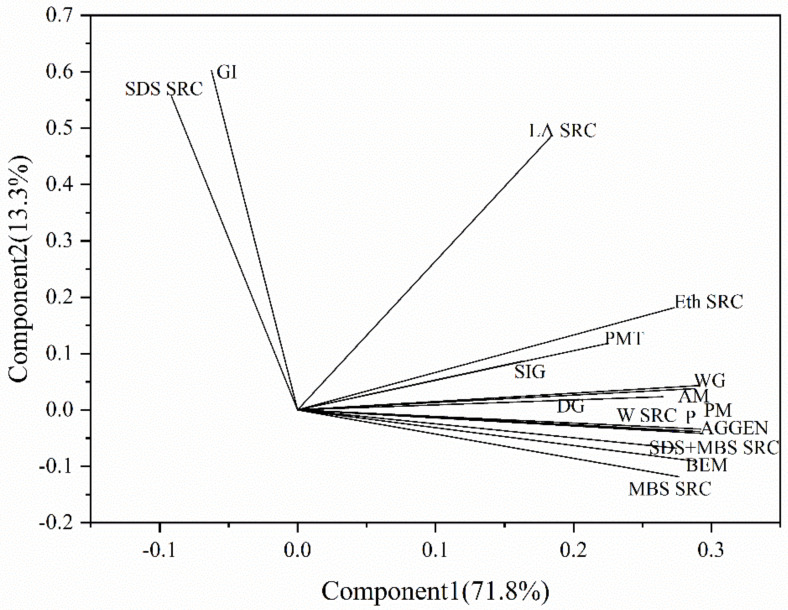
Principal component analysis. P: protein, WG: wet gluten, DG: dry gluten, GI: gluten index, SIG: swelling index of glutenin values, SDSSRC: sodium dodecyl sulphate-SRC, LASRC: lactic acid-SRC, EthSRC: ethanol-SRC, MBSSRC: metabisulfite-SRC, MBS + SDSSRC: sodium dodecylsulphate + metabisulfite-SRC, WSRC: water-SRC, PMT: peak maximum time, BEM: torque maximum, AM: torque 15 s before maximum torque, PM: torque 15 s after maximum torque, AGGEN: gluten aggregation energy.

**Table 1 foods-12-01879-t001:** Flour quality features of ten samples of wheat flour.

Samples	Protein (%)	Wet Gluten (%)	Dry Gluten (%)	Gluten Index (%)	SIG (%)	SDS-Sedi (mL)	SH-Groups (umol/g)	SS-Bonds (umol/g)
1	13.08 ± 0.15 ^a^	38.57 ± 0.02 ^a^	14.01 ± 0.03 ^a^	92.69 ± 0.52 ^a^	6.49 ± 0.06 ^a^	53.05 ± 0.98 ^a^	4.38 ± 0.07 ^b^	2.58 ± 0.04 ^e^
2	11.48 ± 0.04 ^b^	30.84 ± 0.45 ^b^	11.49 ± 0.06 ^b^	95.70 ± 0.06 ^a^	6.09 ± 0.01 ^b^	51.10 ± 1.02 ^b^	4.33 ± 0.14 ^b^	3.64 ± 0.13 ^b^
3	11.03 ± 0.04 ^c^	26.66 ± 0.17 ^d^	10.43 ± 0.15 ^c^	81.27 ± 0.47 ^b^	5.73 ± 0.11 ^c^	46.87 ± 0.36 ^c^	2.94 ± 0.06 ^e^	3.66 ± 0.18 ^b^
4	10.63 ± 0.13 ^d^	26.47 ± 1.09 ^d^	10.52 ± 0.23 ^c^	82.29 ± 0.12 ^b^	5.23 ± 0.05 ^d^	39.38 ± 0.85 ^e^	3.46 ± 0.11 ^d^	4.04 ± 0.12 ^a^
5	10.23 ± 0.08 ^e^	29.42 ± 0.48 ^bc^	9.92 ± 0.01 ^de^	70.92 ± 2.19 ^d^	4.97 ± 0.01 ^e^	45.66 ± 0.11 ^c^	4.50 ± 0.33 ^ab^	3.00 ± 0.20 ^d^
6	10.08 ± 0.08 ^ef^	27.71 ± 0.05 ^cd^	10.23 ± 0.03 ^cd^	65.40 ± 2.52 ^e^	4.88 ± 0.00 ^e^	37.38 ± 0.12 ^f^	4.68 ± 0.03 ^a^	2.60 ± 0.39 ^d^
7	10.05 ± 0.05 ^f^	30.21 ± 0.93 ^b^	10.25 ± 0.19 ^cd^	76.06 ± 1.15 ^c^	4.90 ± 0.03 ^e^	41.43 ± 0.07 ^d^	4.47 ± 0.04 ^ab^	3.07 ± 0.13 ^d^
8	10.03 ± 0.13 ^f^	24.48 ± 0.07 ^e^	9.60 ± 0.06 ^e^	94.03 ± 0.65 ^a^	5.28 ± 0.07 ^d^	51.08 ± 0.58 ^b^	4.28 ± 0.15 ^bc^	3.33 ± 0.07 ^c^
9	5.73 ± 0.08 ^g^	21.50 ± 0.61 ^f^	7.44 ± 0.05 ^f^	94.28 ± 0.53 ^a^	4.34 ± 0.01 ^f^	41.98 ± 0.25 ^d^	4.07 ± 0.14 ^c^	3.23 ± 0.14 ^cd^
10	5.40 ± 0.00 ^h^	19.56 ± 0.33 ^g^	6.58 ± 0.31 ^g^	96.46 ± 0.90 ^a^	3.92 ± 0.04 ^g^	34.60 ± 0.25 ^g^	4.65 ± 0.13 ^a^	3.06 ± 0.19 ^e^

Means with different superscripts within the same column are significantly different at *p* < 0.05. SIG: Swelling index of glutenin; SDS-Sedi: SDS: sedimentation index; 3.3. Analysis of gluten aggregation characteristics.

**Table 2 foods-12-01879-t002:** Mean values of GlutoPeak parameters of ten wheat flour samples.

Samples	PMT (S)	BEM (GPU)	AM (GPU)	PM (GPU)	AGGEN (cm^2^)
1	95.25 ± 3.03 ^d^	64.50 ± 0.87 ^a^	27.25 ± 0.83 ^a^	54.25 ± 1.48 ^a^	1689.25 ± 16.60 ^a^
2	147.25 ± 1.09 ^b^	53.25 ± 0.83 ^b^	20.75 ± 0.43 ^b^	42.75 ± 0.43 ^b^	1351.06 ± 8.09 ^b^
3	94.00 ± 2.55 ^d^	52.25 ± 0.43 ^b^	15.00 ± 0.00 ^d^	38.75 ± 0.83 ^c^	1191.46 ± 16.60 ^c^
4	76.25 ± 1.79 ^g^	49.50 ± 0.87 ^c^	17.75 ± 1.30 ^c^	37.00 ± 0.71 ^d^	1144.93 ± 15.03 ^d^
5	107.00 ± 3.39 ^c^	44.25 ± 1.5 ^d^	15.00 ± 2.35 ^d^	35.25 ± 0.43 ^e^	1082.20 ± 42.34 ^e^
6	81.75 ± 1.48 ^f^	42.75 ± 0.8 ^d^	16.00 ± 1.22 ^cd^	34.25 ± 0.43 ^e^	1057.78 ± 9.29 ^ef^
7	88.50 ± 1.76 ^e^	43.25 ± 2.28 ^d^	12.25 ± 1.09 ^e^	34.00 ± 0.00 ^e^	1012.59 ± 50.00 ^g^
8	156.00 ± 2.12 ^a^	43.00 ± 0.71 ^d^	16.50 ± 1.12 ^cd^	34.50 ± 0.50 ^e^	1036.16 ± 21.31 ^fg^
9	——	——	——	——	——
10	——	——	——	——	——

Means with different superscripts within the same column are significantly different at *p* < 0.05. ——: GlutoPeak parameters were not obtained for this sample. PMT: peak maximum time, BEM: maximum torque, AM: torque 15 s before maximum torque, PM: torque 15 s after maximum torque, AGGEN: gluten aggregation energy.

**Table 3 foods-12-01879-t003:** Solvent retention capacity of gluten in wheat flour.

Samples	Main-SRCs		Supplementary SRCs			
	WSRC (%)	LASRC (%)	EthSRC (%)	SDSSRC (%)	MBSSRC (%)	SDS + MBSSRC (%)
1	76.96 ± 0.46 ^a^	139.97 ± 1.02 ^a^	67.32 ± 0.72 ^a^	92.09 ± 3.58 ^cd^	73.73 ± 0.73 ^a^	74.94 ± 0.14 ^a^
2	66.74 ± 0.42 ^c^	120.56 ± 1.50 ^b^	62.07 ± 0.20 ^ab^	101.66 ± 2.65 ^b^	71.76 ± 2.24 ^a^	70.83 ± 1.06 ^b^
3	69.78 ± 0.20 ^b^	91.50 ± 0.27 ^e^	53.58 ± 0.32 ^c^	83.56 ± 0.81 ^d^	69.44 ± 0.19 ^ab^	66.87 ± 0.22 ^d^
4	68.84 ± 0.49 ^b^	86.48 ± 2.19 ^f^	61.86 ± 1.07 ^ab^	77.01 ± 1.15 ^e^	71.14 ± 0.17 ^a^	69.00 ± 0.50 ^c^
5	64.31 ± 0.58 ^d^	92.88 ± 0.12 ^e^	62.60 ± 1.00 ^a^	89.79 ± 2.39 ^c^	59.92 ± 9.28 ^bcd^	64.91 ± 0.29 ^e^
6	68.92 ± 0.52 ^b^	80.68 ± 1.68 ^g^	63.02 ± 0.77 ^a^	87.25 ± 2.33 ^cd^	66.70 ± 0.70 ^abc^	69.77 ± 1.08 ^ab^
7	63.77 ± 0.16 ^d^	88.44 ± 0.20 ^f^	61.07 ± 0.20 a^b^	84.65 ± 1.33 ^d^	64.88 ± 0.10 ^abc^	63.95 ± 0.09 ^ef^
8	66.63 ± 0.95 ^c^	107.31 ± 1.03 ^c^	61.78 ± 0.09 ^ab^	106.76 ± 3.52 ^a^	61.09 ± 1.59 ^bcd^	63.03 ± 0.19 ^f^
9	54.56 ± 1.21 ^f^	96.68 ± 0.57 ^d^	53.50 ± 0.09 ^c^	96.98 ± 1.27 ^b^	57.70 ± 9.66 ^cd^	60.21 ± 1.41 ^g^
10	58.48 ± 0.26 ^e^	80.18 ± 0.65 ^g^	55.62 ± 9.20 ^bc^	100.18 ± 1.57 ^b^	52.13 ± 0.11 ^d^	53.75 ± 0.50 ^h^

Means with different superscripts within the same column are significantly different at *p* < 0.05. SDSSRC: sodium dodecyl sulphate-SRC, LASRC: lactic acid-SRC, EthSRC: ethanol-SRC, MBSSRC: metabisulfite-SRC, MBS + SDSSRC: sodium dodecyl sulphate + metabisulfite-SRC, WSRC: water-SRC.

**Table 4 foods-12-01879-t004:** Pearson’s r between SRC-values and flour aggregation characteristics and some flour quality traits.

Samples	PMT	BEM	AM	PM	AGGEN	WSRC	LASRC	MBSSRC	MBS + SDSSRC	EthSRC	SDSSRC
PMT	1										
BEM	−0.062	1									
AM	0.132	0.878 **	1								
PM	0.019	0.971 **	0.936 **	1							
GEENE	0.024	0.972 **	0.944 **	0.997 **	1						
WSRC	−0.248	0.841 **	0.827 *	0.838 **	0.919 **	1					
LASRC	0.461	0.791 *	0.877 **	0.879 **	0.584	0.521	1				
MBSSRC	-0.270	0.811 *	0.679	0.725 *	0.880 **	0.833 **	0.516	1			
MBS + SDSSRC	−0.236	0.818 **	0.861 **	0.831 **	0.912 **	0.856 **	0.586	0.941 **	1		
EthSRC	0.040	0.264	0.615	0.447	0.735 *	0.687 *	0.517	0.526	0.690	1	
SDSSRC	0.942 **	0.005	0.276	0.146	−0.292	−0.325	0.375	−0.441	−0.354	−0.061	1
Protein	−0.012	0.989 **	0.914 **	0.994 **	0.999 **	0.919 **	0.569	0.711 *	−0.314	0.885 **	0.906 **
Dry gluten	−0.105	0.934 **	0.915 **	0.978 **	0.960 **	0.908 **	0.709 *	0.898 **	0.941 **	0.752 *	−0.270
Wet gluten	−0.154	0.761 *	0.744 *	0.855 **	0.879 **	0.792 **	0.701 *	0.767 **	0.854 **	0.778 *	−0.244
SIG	0.265	0.937 **	0.858 **	0.923 **	0.916 **	0.846 **	0.791 *	0.881 **	0.874 **	0.557	−0.086

*: Correlation is significant at the 5% level. **: Correlation is significant at the 1% level. SIG: swelling index of glutenin values, SDSSRC: sodium dodecyl sulphate-SRC, LASRC: lactic acid-SRC, EthSRC: ethanol-SRC, MBSSRC: metabisulfite-SRC, MBS + SDSSRC: sodium dodecyl sulphate + metabisulfite-SRC, WSRC: water-SRC, PMT: peak maximum time, BEM: torque maximum, AM: torque 15 s before maximum torque, PM: torque 15 s after maximum torque, AGGEN: gluten aggregation energy.

## Data Availability

Data is contained within the article.

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
