# Peer review of "Correlation Analysis between Wheat Flour Solvent Retention Capacity and Gluten Aggregation Characteristics"

_foods, 2023, doi:10.3390/foods12091879_

Round 1

Reviewer 1 Report

This presented manuscript is dedicated to the investigation of the quality and ability of gluten aggregation of ten types of commercial wheat flour with special emphasis on their correlation with the retention of standards and solvents. The paper sufficiently reflects the aim, methodology and provides interesting theoretical findings, but some shortcomings must be corrected:

Line 43, Line 78-71, All standard methods should be included in the literature list, and properly referenced here.

Line 76, Why were 10 different types of flour selected, how were the samples marked, why were the producers not listed and how were they later differentiated?

Line 78-71, Expand the description of these methods.

Line 140-158, Refer also to the results of other authors in the available literature and compare with those obtained for protein content and swelling index.

Line 157-158 Reference is missing.

Line 229-232, Better explain the meaning and significance of LASRC and WSRC values.

Line 263, It means table 4 not figures 4.

Line 285, SDS, SRC and GI

Write all abbreviations and their meanings below the tables and figures.

Figure 2, Why are the samples not included in the PCA analysis; Instead of the component on the axes write PC 1 and PC 2;  What is the influence of the variables on the coordinates PC1 and PC2 (this part is missing)?

Reviewer 2 Report

This manuscript investigates how solvent retention capacity is related to different wheat flour characteristics by measuring the properties of different commercial brands of wheat flour. One interesting outcome of this study is the use of new solvents that have been demonstrated to provide new insight into the prediction of wheat flour quality. This new knowledge could be valuable to the baking industry, allowing them to make better-informed decisions when selecting flour for their products.

In spite of this, there are some remarks that need to be addressed.

First, I found the article extremely difficult to read because it contained so many acronyms, some of which are not even explained. I had to constantly pause to look up the whole meaning of each acronym, which disrupted the overall reading experience. For example, what is DTNB? It would be helpful to insert a table containing all the acronyms used in the text.

English requires some attention. Most sentences are written in the past tense. However, there are also many sentences written in the infinite form. (e.g., rows 90-91, 98-99, 108, 109, 116, 117, 120-121, 123, 176 etc.). The infinite form should generally be avoided as it can make the sentence unclear and lead to misunderstanding.

How many replicates were carried out for each measurement? This detail should be reported.

There is a limited amount of information provided in the materials section. What are the main differences between the different brands? Is there any particular reason for their choice?

I suggest providing a more detailed explanation of the outcomes of Table 4. The table should be at the core of the manuscript, as it is essential to understand the implications of the study and its results. A thorough explanation of the outcomes will also help readers better understand the implications of the data.

Minor details

Table 1: The first column's name, "Categoris" does not mean anything to me. I suppose that they are the samples.

Figure 1: Write the labels of the figure in English.

Table 2: Why data on two samples are missing?

Figure 2: This is not exactly a PCA plot. It is the plot of the loadings. The quality and the readability of the figure must be improved.

The reference section does not appear to be in accordance with the journal's format.

Round 2

Reviewer 2 Report

Reviewers' suggestions have been taken into account by the authors.

It should be noted, however, that English is still sloppy, even in the new sentences that have been added. I find this a little annoying.

I strongly recommend that the article be proofread more accurately before final acceptance.
